

# Designing conservation strategies to preserve the genetic diversity of *Astragalus edulis* Bunge, an endangered species from western Mediterranean region

Julio Peñas[1], Sara Barrios[2], Javier Bobo-Pinilla[2,3], Juan Lorite[1] and M. Montserrat Martínez-Ortega[2,3]

[1] Plant Conservation Unit, Department of Botany, University of Granada, Granada, Spain
[2] Department of Botany, University of Salamanca, Salamanca, Spain
[3] Biobanco de ADN Vegetal, Edificio Multiusos I+D+i, Calle Espejo s/n, Salamanca, Spain

## ABSTRACT

*Astragalus edulis* (Fabaceae) is an endangered annual species from the western Mediterranean region that colonized the SE Iberian Peninsula, NE and SW Morocco, and the easternmost Macaronesian islands (Lanzarote and Fuerteventura). Although in Spain some conservation measures have been adopted, it is still necessary to develop an appropriate management plan to preserve genetic diversity across the entire distribution area of the species. Our main objective was to use population genetics as well as ecological and phylogeographic data to select Relevant Genetic Units for Conservation (RGUCs) as the first step in designing conservation plans for *A. edulis*. We identified six RGUCs for in situ conservation, based on estimations of population genetic structure and probabilities of loss of rare alleles. Additionally, further population parameters, i.e. occupation area, population size, vulnerability, legal status of the population areas, and the historical haplotype distribution, were considered in order to establish which populations deserve conservation priority. Three populations from the Iberian Peninsula, two from Morocco, and one from the Canary Islands represent the total genetic diversity of the species and the rarest allelic variation. Ex situ conservation is recommended to complement the preservation of *A. edulis*, given that effective in situ population protection is not feasible in all cases. The consideration of complementary phylogeographic and ecological data is useful for management efforts to preserve the evolutionary potential of the species.

Corresponding author
Julio Peñas, jgiles@ugr.es

## INTRODUCTION

Although one of the central concepts in biodiversity conservation is that genetic diversity is crucial to ensure the survival of species, until now the conservation of plant genetic resources has received less attention than it deserves. Plant-conservation strategies have been commonly based on general premises, leading to more or less standardized systems for evaluating the extinction risks of the species (*Moraes et al., 2014*). However, plant species differ enormously in biological traits and environmental requirements, making it

unrealistic to apply a single system to all species. Recent years have seen increasing efforts to improve both in situ and ex situ conservation methods, which in theory would foster dynamic conservation of plant species and populations (*Volis & Blecher, 2010*; *Heywood, 2014*). Plant genetic diversity is spatially structured at different scales (e.g. geographical areas, populations, or among neighbouring individuals) (*Engelhardt, Lloyd & Neel, 2014*) as a result of environmental influences, life-history traits, and the demographic past history of the species. Therefore, management schemes for conservation often require an understanding of population dynamics and knowledge of relative levels of genetic diversity, within species genetic structure, as well as within- and among-population genetic differentiation in order to focus efforts on specific populations needing recovery (*Haig, 1998*; *Pérez-Collazos, Segarra-Moragues & Catalán, 2008*).

Several estimators have been assayed to answer the question of which and how many populations deserve conservation priority, such as: Evolutionary Significant Units (ESUs; *Ryder, 1986*); Management Units (MUs; *Moritz, 1994*); Operational Conservation Units (OCUs; *Doadrio, Perdices & Machordom, 1996*); Fundamental Geographic and Evolutionary Units (FGEUs; *Riddler & Hafner, 1999*); Functional Conservation Units (FCUs; *Maes et al., 2004*), among others (see also *Pérez-Collazos, Segarra-Moragues & Catalán, 2008*; *Domínguez-Domínguez & Vázquez-Domínguez, 2009*). *Fraser & Bernatchez (2001)* reviewed the different concepts of ESUs (the most prominent estimator among those previously mentioned), concluding that differing criteria would work more dynamically than others and can be used alone or in combination depending on the situation. *Pérez-Collazos, Segarra-Moragues & Catalán (2008)*, partially based on *Caujapé-Castells & Pedrola-Monfort (2004)*, as well as on the premises established by *Ciofi et al. (1999)*, introduced the concept of Relevant Genetic Units for Conservation (RGUCs), which was subsequently used to propose sampling strategies for species such as *Boleum asperum* Desv. (*Pérez-Collazos, Segarra-Moragues & Catalán, 2008*) and *Borderea pyrenaica* Miégev. (*Segarra-Moragues & Catalán, 2010*). This approach combines two methods that use genetic data (considering both usual and rare alleles) to estimate the minimum number of conservation units (often corresponding to populations) that should be targeted for an adequate representation of the total (or partial) genetic variability of a threatened species, as well as a way to select among all units (i.e. populations) which contain a singular or rare allelic composition. A list of preferred sampling areas (PSA) indicating the geographical ranges with higher probabilities of capturing a particular rare allele is finally established, helping to identify RGUCs and therefore prioritize particular populations, as well as sampling for ex situ conservation. This method helps identify the most singular populations, based on the idea that rare alleles are essential in conservation because they represent unique evolutionary products that could provide the species with advantageous properties to cope with eventual environmental shifts. Thus, collection designs oriented to sampling rare alleles reinforce declining populations and may aid the survival of reintroduced plants (*Bengtsson, Weibull & Ghatnekar, 1995*; *Pérez-Collazos, Segarra-Moragues & Catalán, 2008*). One of the main advantages of this genetic conservation approach is that it objectively prioritizes particular plant populations in low-extinction-risk categories (*Segarra-Moragues & Catalán, 2010*),

particularly in taxa that have many populations and individuals, making active protection and monitoring of the entire distribution area of the species difficult or unaffordable.

The species selected for this study *Astragalus edulis* Bunge (Fabaceae), is an annual plant that inhabits semidesertic areas of south-eastern Spain, western North Africa, and the Canary Islands (Fuerteventura and Lanzarote) (*Peñas, 2004*; *Reyes-Betancort et al., 2005*). It is a threatened species evaluated as Endangered (EN) in Spain. Despite its relatively wide distribution area, only a few populations remain, these being highly fragmented. Habitat alteration has been cited as a major threat to this species (*Peñas, 2004*). Specifically, the abandonment of traditional agricultural practices, overgrazing, and the habitat depletion, caused by the spread of greenhouses, may have had severely negative consequences for species survival (*Benito et al., 2009*). This species represents an ideal model to test the utility of RGUC identification as an affordable way to conserve taxa that have highly fragmented populations, some of them with many individuals, but they are under extinction-risk categories.

Our specific aims are: (1) to evaluate the distribution of the genetic diversity among the different populations, and/or geographical areas; (2) to assess the number of populations that should be sampled or preserved in order to establish a representative percentage of the total genetic variation of *A. edulis*; (3) to identify which populations should be prioritized to better represent the genetic singularity and geographic variability for both ex situ and in situ conservation.

## MATERIALS AND METHODS

### Studied species

*Astragalus edulis* Bunge (Fabaceae) is a short-lived therophytic, hermaphroditic plant. Until now, no information has been available on population sizes, except for the rough estimates by *Peñas (2004)*, indicating that ca. 226,000 individuals were present in SE Spain in 2003. This estimate also indicated a noticeable inter-annual fluctuation in population sizes (number of individuals) and reproductive success (*Peñas, 2004*; *Reyes-Betancort et al., 2005*). The reproductive biology of the species is poorly known; it shows an entomophilous pollination syndrome, lacking asexual reproduction as well as evident adaptations to long-distance dispersal, but there is no information available on its pollination biology or dispersal agents. Its habitat is restricted to grasslands on poor sandy soils, resulting from erosion or deposition of volcanic or schistose rocks in semiarid areas of the western Mediterranean region (*Peñas, 2004*; *Reyes-Betancort et al., 2005*) (Fig. 1).

*Astragalus edulis* is rare (i.e. constantly sparse in a specific habitat but over a large range; according to *Rabinowitz, 1981*) and threatened species evaluated as Endangered (EN) in Spain, and consequently included in the Spanish national and regional red lists (*Bañares et al., 2004*), as well as in the Andalusian (southern Spain) red list (*Cabezudo et al., 2005*). Also, some populations in Spain are included in Natura 2000 network (Special Areas of Conservation, Council Directive 92/43/EEC) and in Regional Network of Natural Protected areas of Andalusia (southern Spain), while the areas occupied by the species in Canary Islands and Morocco lack legal protection.
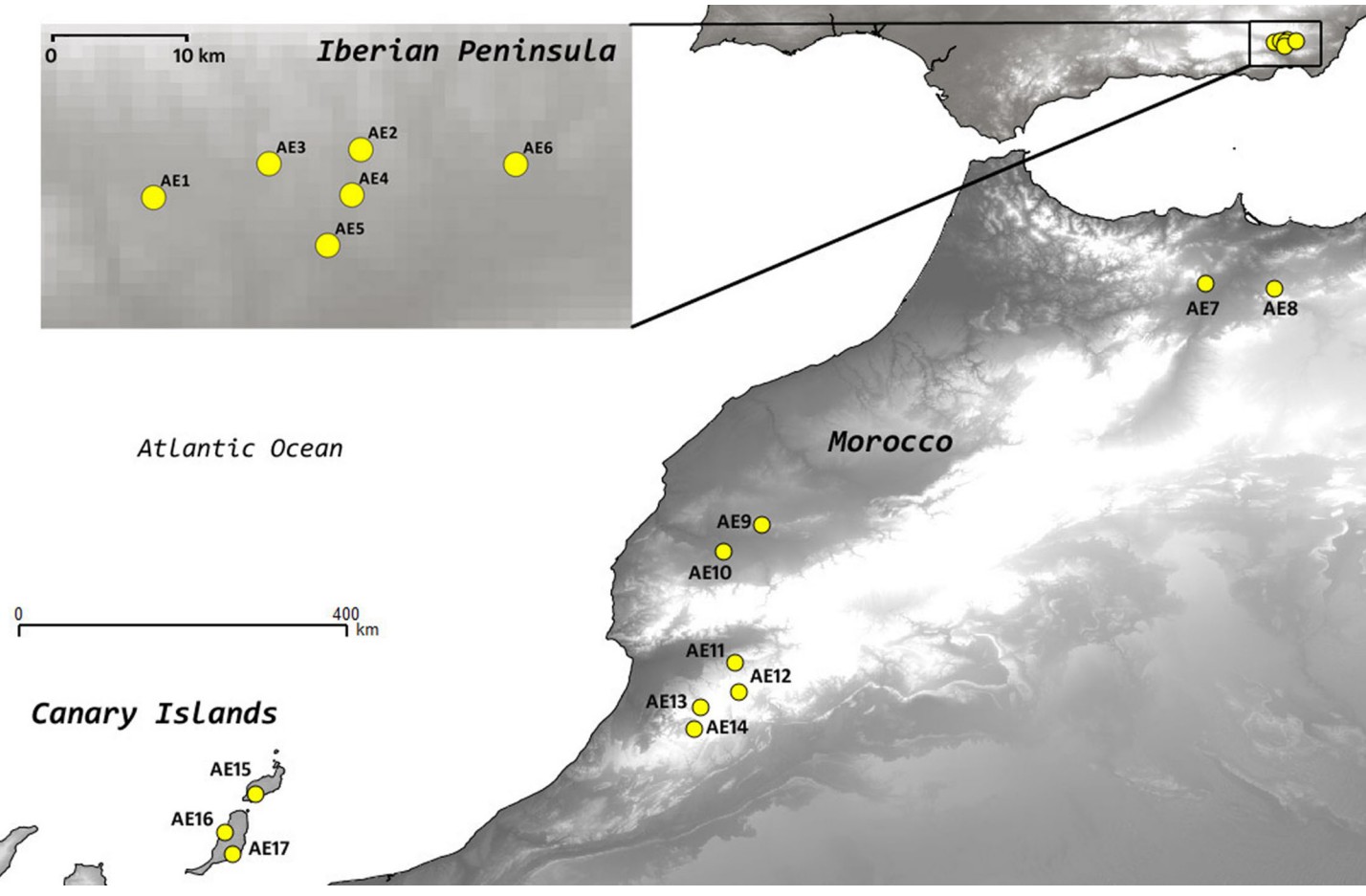

**Figure 1** Location of the populations of *Astragalus edulis* sampled for this study.

## Plant material for DNA study

We collected fresh leaf tissue from 360 individuals belonging to 17 populations; 6 from the Iberian Peninsula (AE1 to AE6), 8 from Morocco (AE7 to AE14) and 3 from the Canary Islands (AE15 to AE17), spanning the entire distribution range of the species (Table 1; Fig. 1). We considered different populations when individual are more than 1 km apart. We aimed to collect 25 individuals per population whenever possible but due to small population sizes in some cases the final number of individuals sampled per population ranged from 7 to 33. Within a particular population the samples were collected at distances greater than 5 m apart to avoid sampling closely related individuals. All sampling sites were geo-referenced with a GPS (GARMIN GPSMAP 60) and vouchers of the sampled localities were included in the herbaria of the Universities of Salamanca (SALA) and Granada (GDA). Plant material from each individual was dried and preserved in silica gel until DNA extraction.

## DNA isolation, AFLP protocol and cpDNA sequencing

Total DNA was isolated following the 2x CTAB protocol (*Doyle & Doyle, 1987*) with minor modifications. AFLP profiles were drawn following established protocols

**Table 1 Geographic features of the populations sampled in the study.** (N) Number of individuals used for the AFLP analyses.

| Population code | Locality | Altitude | Longitude | Latitude | N |
|---|---|---|---|---|---|
| AE1 | Spain; Almería, Alcubillas | 735 | −2.6025 | 37.0987 | 16 |
| AE2 | Spain; Almería, Tabernas | 915 | −2.4643 | 37.1306 | 24 |
| AE3 | Spain; Almería, Gérgal | 720 | −2.5254 | 37.1209 | 32 |
| AE4 | Spain; Almería, Gérgal, Arroyo Verdelecho | 648 | −2.4704 | 37.1002 | 24 |
| AE5 | Spain; Almería, Tabernas, Desierto de Tabernas | 621 | −2.4863 | 37.0668 | 23 |
| AE6 | Spain; Almería, Filabres, Rambla del Saltador | 541 | −2.3610 | 37.1206 | 33 |
| AE7 | Morocco; La Oriental, between El-Aïoun and Tanarchefi | 919 | −2.6016 | 34.4174 | 17 |
| AE8 | Morocco; Taza, Jebel Guilliz | 425 | −3.3496 | 34.4669 | 21 |
| AE9 | Morocco; Marrakech, Chemaia, prox. Kettara | 480 | −8.1875 | 31.8729 | 22 |
| AE10 | Morocco; Marrakech, between Marrakech and Chichaoua | 380 | −8.6185 | 31.5720 | 14 |
| AE11 | Morocco; Taroudant, between Tasgount and Ighil | 1,437 | −8.4832 | 30.1831 | 18 |
| AE12 | Morocco; Taroudant, between Irherm and Tata | 1,710 | −8.4478 | 30.0467 | 19 |
| AE13 | Morocco; Taroudant, Tafraoute, Tizi-n-Tarakatine, prox. El Jebar | 1,484 | −8.8587 | 29.7376 | 25 |
| AE14 | Morocco; Taroudant, between Tafraoute and Tleta-Tasrite | 1,620 | −8.9385 | 29.6354 | 7 |
| AE15 | Spain; Canary Islands; Lanzarote, Vega de Temuime | 159 | −13.728 | 28.9337 | 29 |
| AE16 | Spain; Canary Islands; Fuerteventura, Tiscamanita | 234 | −14.033 | 28.3576 | 14 |
| AE17 | Spain; Canary Islands; Fuerteventura, Barranco de Majada Blanca | 181 | −13.986 | 28.2673 | 22 |

(*Vos et al., 1995*) with modifications. A negative control sample was consistently included to test for contamination, and five samples taken at random were replicated to test for reproducibility. Selective primers were initially screened using 24 primer combinations for the selective PCR and three were finally selected (fluorescent dye in brackets): EcoRI-AGA (6-FAM)/MseI-CTG, EcoRI-AAG(VIC)/MseI-CAG and EcoRI-ACC(NED)/MseI-CTG, because they generated a relatively high number (a high number of alleles per individual is desirable in conservation genetic studies given that AFLP are dominant markers; *Lowe, Harris & Ashton, 2004*) of clearly reproducible bands, for which homology was easy to ensure. The fluorescence-labelled selective amplification products were separated in a capillary electrophoresis sequencer (ABI 3730 DNA Analyzer; Applied Biosystems, Foster City, CA, USA), with GenScan ROX (Applied Biosystems, Foster City, CA, USA) as the internal size standard, at the Genomic Department of Universidad Politécnica de Madrid. Raw data with amplified fragments were scored and exported as a presence/absence matrix.

To complement the information of the mainly nuclear AFLPs, the plastid regions *trn*G-*trn*S, *trn*C-*rpo*B, and *tab*F-*tab*C (*Taberlet et al., 1991*; *Shaw et al., 2005*) were explored (see Table 2 for details). These regions showed the highest variability of

**Table 2  PCR primers and conditions used to obtain cpDNA sequence data for *Astragalus edulis*.**

| cpDNA region | Forward primer | Reverse primer | Denaturation Temperature/ Time | Annealing Temperature/ Time | Extension Temperature/ Time | Cycles |
|---|---|---|---|---|---|---|
| *trn*G-*trn*S | 3'trnG$^{UUC}$ | trnS$^{GCU}$ | 95°C/30″ | 62°C/30″ | 72°/1′30″ | 35 |
| *trn*C-*rpo*B | trnC$^{GCA}$R | rpoB | 95°C/30″ | 55°C/30″ | 72°/1′30″ | 35 |
| *tab*C-*tab*F | trnL$^{UAA}$5′ | trnF$^{GAA}$ | 95°C/30″ | 52°C/30″ | 72°/2′30″ | 35 |

23 surveyed cpDNA regions in the preliminary studies using 10 individuals and were therefore used to analyse a total of 61 individuals (i.e., 3–4 individuals per population, due to amplification failure in 7 cases) of *A. edulis* : 38 from Iberian Peninsula (IP), 17 from Morocco (M) and, 6 from Canary Islands (CI). PCR products were purified using PCR Clean-Up with ExoSAP-IT Kit (AFFIMETRIX, Santa Clara, CA, USA) following the manufacturer's instructions. The cleaned amplification products were analysed with a 3,730 DNA Genetic Analyzer capillary sequencer (Applied Biosystems, Foster City, CA, USA). All sequences were deposited in GenBank (see Supplemental Information).

## Molecular data analysis

An unrooted phylogram based on Nei and Li's genetic distances (*Nei & Li, 1979*) and AFLP data was calculated using the Neighbour-Joining (NJ) clustering method, with 1000 bootstrap pseudoreplicates (BS), in order to evaluate genetic structure within *A. edulis*. This was conducted with the software PAUP v4.0b10 (*Swofford, 1998*). As an additional estimate of the population genetic structure and based on Dice's similarity coefficient (*Dice, 1945*; *Lowe, Harris & Ashton, 2004*), a Principal Coordinate Analysis (PCoA) was performed with NTSYS-pc 2.02 (*Rohlf, 2009*) as an additional approach to the overall genetic relationships among the individuals analysed.

An analysis of molecular variance (AMOVA) was performed with the software ARLEQUIN 3.5.1.2 (*Excoffier, Laval & Schneider, 2005*). The analysis was first conducted considering all populations belonging to the same group and, second, partitioning genetic variation into portions assignable to differences among three predefined groups (the three main geographic groups derived from the NJ phylogram, i.e. (IP: AE1–AE6), (M: AE7–AE14), and (CI: AE15–AE17)) in order to test for identifiable genetic structures among geographical divisions. Significance levels of the variance components were estimated for each case using non-parametric permutations with 1023 replicates.

The proportion on polymorphic alleles measured by Nei's gene-diversity index (*Nei, 1987*) was calculated for each population using the R package AFLPDAT for R (*Ehrich, 2006*). This package was also used to calculate the frequency down-weighted marker values per population or sampling site (DW; *Schönswetter & Tribsch, 2005*), which estimates genetic rarity of a population as equivalent to range down-weighted species values in historical biogeographical research (*Crisp et al., 2001*). Finally, the number of rare alleles ($N_r$), (i.e. bands that showed an overall frequency lower than 10%, and that are present in less than 20% of the populations (*Pérez-Collazos, Segarra-Moragues & Catalán, 2008*), was calculated as an additional measure of rarity.

The completeness of haplotype sampling across the range of *A. edulis* was estimated using the Stirling probability distribution. It provides a way to evaluate the assumption that all haplotypes have been sampled (*Dixon, 2006*). Plastid-DNA sequences were assembled and edited using GENEIOUS PRO™ 5.4 (*Drummond et al., 2012*) and aligned with CLUSTAL W2 2.0.11 (*Larkin et al., 2007*), and further adjustments were made by visual inspection. The resulting sequences were concatenated; the gaps longer than one base pair were coded as single-step mutations and treated as a fifth character state. An unrooted haplotype network was constructed using the statistical parsimony algorithm (*Templeton, Crandall & Sing, 1992*) as implemented in TCS 1.21 (*Clement, Posada & Crandall, 2000*), and used to infer the existing genealogical relationships.

## Selection of relevant genetic units for conservation (RGUCs)

The selection of RGUCs is based on AFLP data and relies on the combination of two methods based on population structure and probabilities of loss of rare alleles.
In summary, the values of the probability of rare-allele loss are compared with those of the degree of inter-population subdivision (*Caujapé-Castells & Pedrola-Monfort, 2004*; *Pérez-Collazos, Segarra-Moragues & Catalán, 2008*).

First, the population-differentiation coefficient ($F_{ST}$) obtained with ARLEQUIN was used to estimate the total number of populations that should be targeted, according to the *Ceska, Affolter & Hamrick (1997)* equation modified $P = 1 - F_{ST}^n$ (*Segarra-Moragues & Catalán, 2010*; but not *Pérez-Collazos, Segarra-Moragues & Catalán, 2008*) where $n$ is the number of populations to be sampled to represent a given proportion ($P$) of the among-population genetic diversity. For *A. edulis*, a $P$ value of 99.9% of the total genetic diversity was established, to cope properly with high conservation standards.

Second, using the mean frequencies of rare bands (i.e. with an overall frequency lower than 10% and present in less than 20% of the populations) and their associated probabilities of loss, the probability that a sample size on $N$ populations fails to include an allele with population frequency $p$ was calculated (*Caujapé-Castells & Pedrola-Monfort, 2004*; *Pérez-Collazos, Segarra-Moragues & Catalán, 2008*). For this, the expression $L ¼ = (1 - p)^{2N}$ (*Bengtsson, Weibull & Ghatnekar, 1995*) was used, where $p$ represents the allele frequency and $N$ the number of populations in which a rare allele is present (*Pérez-Collazos, Segarra-Moragues & Catalán, 2008*). For each rare allele, the observed ($Lo$) and expected ($Le$) probabilities of loss were calculated. The negative natural logarithms ($-Log\ Lo$ and $-Log\ Le$) of those values were plotted (y-axis) against the mean frequency of each rare allele (x-axis) and used to calculate the respective linear regressions. The representative $R$ value (which indicates the proportion of rare alleles captured by sampling only one population) was calculated as the quotient between the slope of the expected regression line and the slope of the observed regression line, i.e. $R = m(-Log\ L_e)/m(-Log\ L_o)$ (*Bengtsson, Weibull & Ghatnekar, 1995*; *Caujapé-Castells & Pedrola-Monfort, 2004*; *Pérez-Collazos, Segarra-Moragues & Catalán, 2008*; *Segarra-Moragues & Catalán, 2010*).

Several qualitative features of the populations and habitat disturbances were recorded during the field work in order to combine them with the measures of genetic diversity. For this, we selected population variables that were accounted as follows (adapted from

*IUCN, 2001*): i) Occupation area: small <1 km$^2$ vs. large >1 km$^2$, ii) population size: high >1,000 individuals vs. low <1,000 individuals), iii) vulnerability: stable = with no disturbances or with minor disturbances/declining = with clear disturbance of both individuals and habitat/critically declining = major disturbances, with major disturbance of individuals and habitat; and iv) conservation status of the area: protected vs. unprotected.

Generalized linear models were used to test whether the main genetic diversity and rarity parameters (i.e. $h_{Nei}$, DW, and Nr) show associations with qualitative population and conservation features. Beforehand, to enhance the robustness of the models, we resampled the cases 10,000 times by bootstrapping using the R boot package (*Canty & Ripley, 2013*). Nei's diversity index and the frequency of down-weighted marker values were fitted to Gaussian distributions, whereas the number of rare alleles was fitted to a Poisson distribution. To test significant level differences of a given variable, we used the glht function of the R multcomp package, indicated for multiple comparisons in generalized linear models (*Hothorn, Bretz & Westfall, 2008*).

## RESULTS

### Genetic variability and structure

A total of 1134 reliable polymorphic bands (averaging ca. 45 per individual per primer combination) were found from the three primer pairs selected for the 360 individuals studied. The final error rate was insignificant (1.67%). The number of rare alleles, DW values and Nei's genetic diversity values corresponding to each population are given in Table 3. AFLPs detected low levels of intrapopulation genetic diversity for *A. edulis*. Nei's gene diversity index ranged from a minimum value of 0.066 (AE7; in the easternmost population of Morocco) to a maximum of 0.155 (AE5; in the central part of the Iberian distribution of the species) and the diversity values were similar across all other populations studied. The total species diversity was 0.108. Regarding rarity, the genetically most distinctive population (DW = 5.713) appeared to be AE16 in Fuerteventura, while the lowest DW values were found in the easternmost part of the Iberian core (AE6; DW = 1.507).

Both the unrooted NJ tree and the PCoA based on the entire data set (Fig. 2) revealed well-defined genetic structure of populations in correspondence to geographic groups. The first group (Fig. 2A) includes all populations from the Iberian Peninsula (85% BS), a second cluster those from Morocco (74% BS) and the third those from the Canary Islands (100% BS), plus some individuals from Morocco (two samples from AE9), although the relationship between these latter two groups is weak (62% BS) and the Moroccan part of this cluster seems to be closely related to the remaining Moroccan individuals. The same geographical groups are revealed by the PCoA (Fig. 2B), but in this case the apparently close relationship between some of the Moroccan and all the Canarian samples suggested by NJ does not seem to be supported, while an affinity between the Moroccan and the Iberian individuals is suggested. The first three axes account for 13.2, 6.4, and 4.7% of the total variance, respectively.

**Table 3 Population, geographical groups, AFLP derived diversity and rarity descriptors, rarity assessment through qualitative variables (see text) and cpDNA haplotypes (endemic ones in bold characters) for the studied population of _A. edulis_.** Geographical groups: IP, Iberian Peninsula; M, Morocco; CI, Canary Islands; $h_{Nei}$, Nei's diversity index (_Nei 1987_); DW, frequency down-weighted marker values; $N_r$, number of rare alleles; H, haplotype.

| Population | Geographical group | $h_{Nei}$ | DW | $N_r$ | Occupation area | Population size | Vulnerability | Legal status | H |
|---|---|---|---|---|---|---|---|---|---|
| AE1 | IP | 0.101 | 3.505 | 31 | small | reduced | critical | unprotected | IV,**V** |
| AE2 | IP | 0.103 | 2.226 | 25 | large | high | moderate | protected | I,**V** |
| AE3 | IP | 0.125 | 3.298 | 45 | large | high | moderate | protected | I,IV |
| AE4 | IP | 0.151 | 4.038 | 38 | large | high | acceptable | protected | I,**III** |
| AE5 | IP | 0.155 | 4.644 | 47 | large | high | acceptable | protected | I,IV,**V** |
| AE6 | IP | 0.076 | 1.507 | 16 | large | reduced | moderate | unprotected | I |
| AE7 | M | 0.066 | 1.754 | 14 | small | reduced | critical | unprotected | I |
| AE8 | M | 0.119 | 3.2 | 33 | large | high | moderate | unprotected | I |
| AE9 | M | 0.114 | 3.218 | 51 | small | reduced | critical | unprotected | IV |
| AE10 | M | 0.082 | 1.728 | 8 | small | reduced | moderate | unprotected | **VI** |
| AE11 | M | 0.104 | 2.924 | 27 | large | reduced | moderate | unprotected | **II** |
| AE12 | M | 0.097 | 2.834 | 30 | small | reduced | critical | unprotected | IV |
| AE13 | M | 0.103 | 2.815 | 33 | large | high | moderate | unprotected | IV |
| AE14 | M | 0.076 | 2.08 | 12 | small | reduced | critical | unprotected | IV |
| AE15 | CI | 0.074 | 2.862 | 14 | small | high | moderate | unprotected | **VII** |
| AE16 | CI | 0.127 | 5.713 | 37 | small | reduced | moderate | unprotected | **VII** |
| AE17 | CI | 0.110 | 4.996 | 55 | large | reduced | acceptable | unprotected | **VII** |

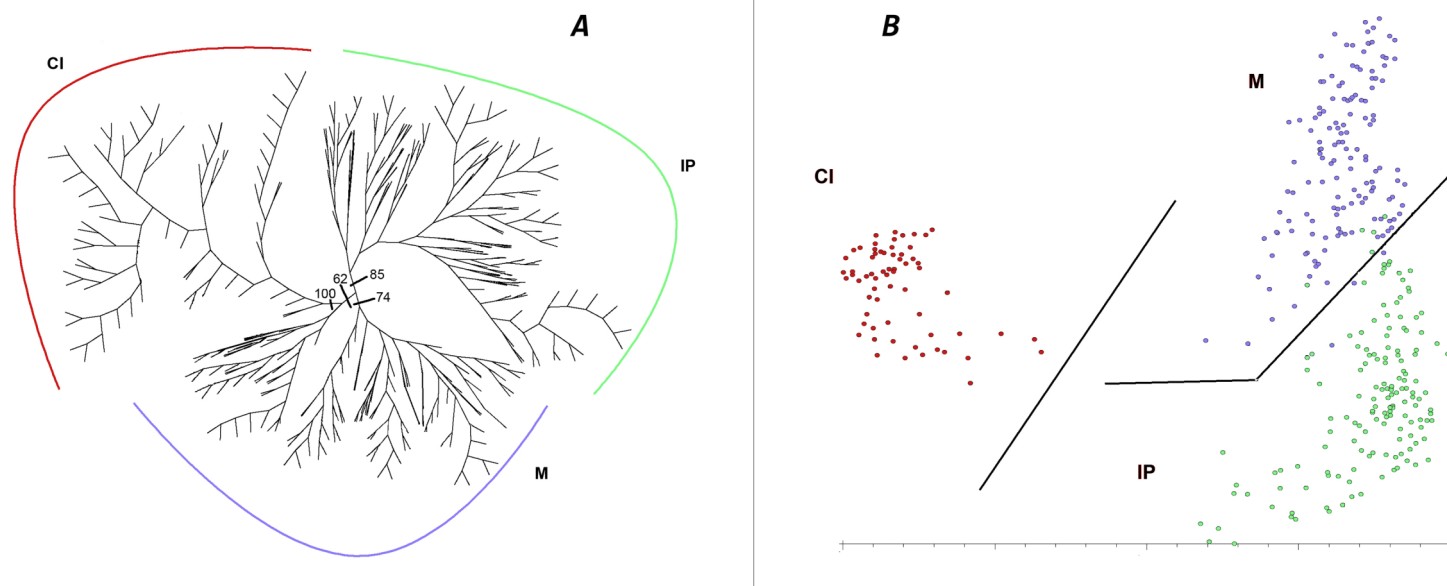

**Figure 2 Cluster analysis of genetic diversity, using AFLPs, in _Astragalus edulis_.** (A) Neighbour-Joining analysis, BS values are indicated; (B) PCoA. Geographical groups: IP, Iberian Peninsula: M, Morocco: CI, Canary Islands.

Table 4 Comparison of analyses of molecular variance (AMOVA), based on AFLP data, of *Astragalus edulis* across the main geographical groups (IP, Iberian Peninsula; M, Morocco; CI, Canary Islands), and populations (are shown in brackets) (see Table 1 and Fig. 1).

| Source of variation | MS | d.f. | Absolute variation | Percentage of variation | $F_{ST}$ | 95% confidence interval |
|---|---|---|---|---|---|---|
| **One group (A1–A17)** | | | | | 0.289 | 26.2–30.8 |
| Among populations | 9268.217 | 16 | 24.641 | 28.94 | | |
| Within populations | 20755.722 | 343 | 60.512 | 71.06 | | |
| **Three groups: IP(A1–A6); M(A7–A14) and C(A15–A17)** | | | | | 0.346 | 21.1–26.8 |
| Among groups | 5694.211 | 2 | 22.611 | 24.44 | | |
| Among populations | 3574.006 | 14 | 9.383 | 10.14 | | |
| Within populations | 20755.722 | 343 | 60.512 | 65.41 | | |

AMOVA analysis of the entire data set as a single group (Table 4) revealed that the genetic variation among individuals (71.06%) is meaningfully higher than the variation among populations (28.94%, $F_{ST} = 0.289$, $p < 0.001$). The results of a hierarchical AMOVA confirm that a population division into the three geographic groups defined by NJ and PCoA analyses reveals 24.44% of the variance attributed to differences among these geographical areas ($F_{ST} = 0.346$, $p < 0.001$), while only 10.14% of the variance is attributed to differences among populations within these three geographic groups.

The length of the three cpDNA regions for 61 individuals was 712 to 926 bp, and resulted in an alignment of 2545 bp (2549 characters with indels coded). The genetic variability within *A. edulis* was remarkably low (26 cpDNA regions initially tested, 3 of them used to analize a total of 61 individuals), and all the mutations together defined a total of 7 haplotypes. The completeness of haplotype sampling estimated using *Dixon's (2006)* method was 0.95 (most likely value of haplotypes = 7.002), suggesting that all haplotypes present in the species were sampled. TCS implied a 95% parsimony network with a maximum limit of five steps (Fig. 3). The most frequent haplotype (I) was found in five populations from the Iberian Peninsula and in the north-eastern Moroccan populations, while the second most frequent haplotype (IV) was represented in four western Moroccan populations and also in two Iberian populations. Within the Iberian Peninsula, two endemic haplotypes (III and V) were found and the western Moroccan populations also showed two endemic haplotypes (II and VI). A single endemic haplotype (VII) was found in Fuerteventura and Lanzarote (Fig. 3; Table 3).

## Identification of RGUs

According to our results, 99.9% of the overall genetic diversity through the entire distribution range of *A. edulis* would be represented by just 6 populations (N = 5.69). This should be the minimum number of populations to be targeted for suitable conservation. Of the total 1134 alleles detected by the AFLP analysis, 273 complied with the established rarity criteria (Table 3; Appendix 1). Of these rare alleles, 66 were exclusive to the Iberian Peninsula), 78 to Morocco and 57 to the Canary Islands; the remaining rare bands were

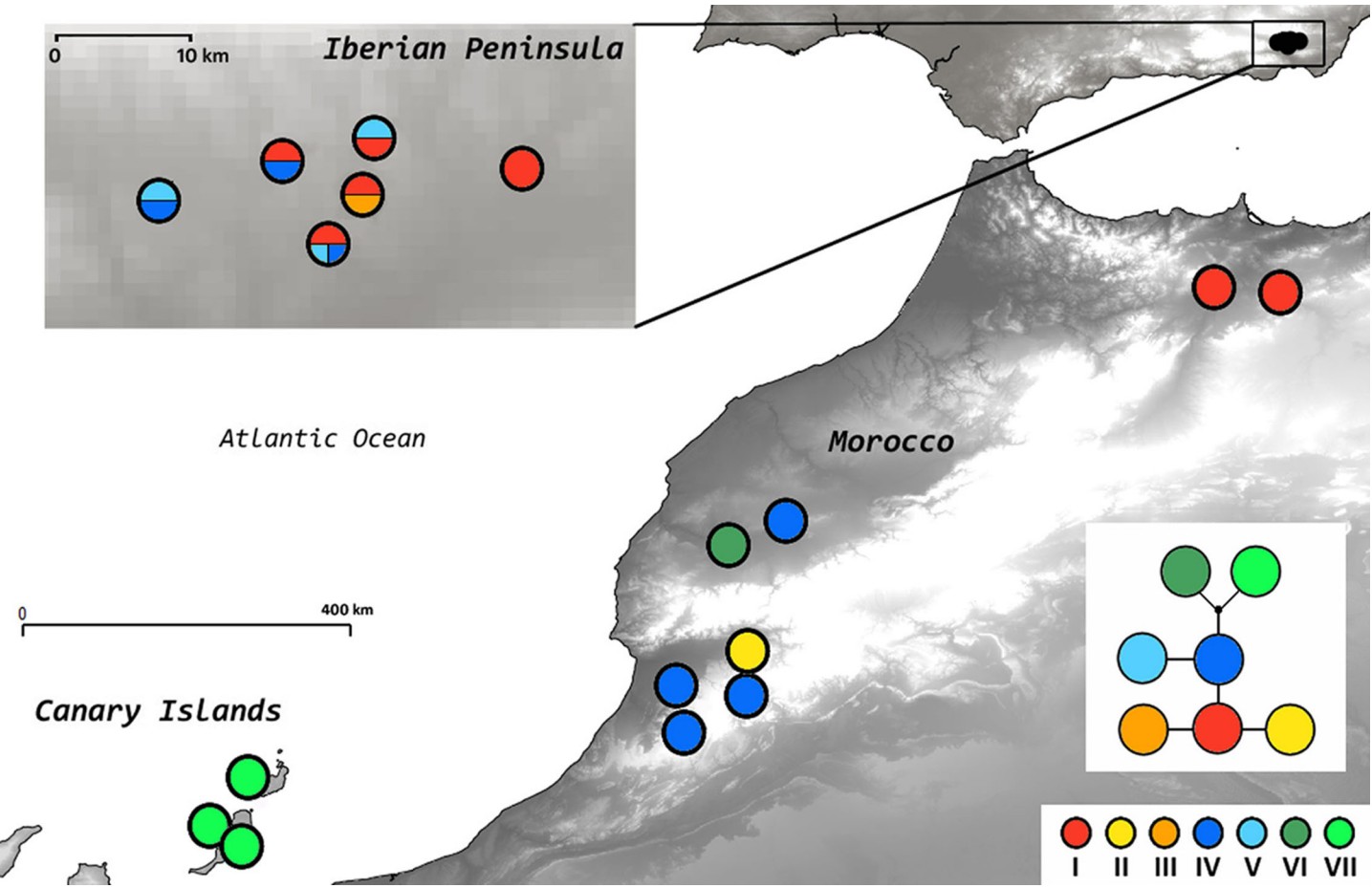

**Figure 3 Statistical parsimony network and geographical distribution of plastid DNA haplotypes.** The insert shows populations within the Iberian Peninsula. The small black dot represents a missing intermediate haplotype. Sectors within circles in the map indicate the presence of different haplotypes in different individuals of the same population.

distributed among different populations of the three geographical regions (detailed data available upon request). The representative R-value (i.e. proportion of rare alleles determined by sampling only one population) considering *A. edulis* as one group was R = 0.354. This means that the sampling of a single population of the entire distribution area of the species would represent the 35.4% of the whole set of rare alleles of the species. This value, calculated independently for each geographic area, showed slight variations (i.e. IP: R = 0.407, M: R = 0.355 and CI: R = 0.293). Based on the mean frequencies of the rare alleles, as well as on their distribution among populations, the areas where each of these alleles had the highest probability of being found by randomly sampling one population were: IP (124), M (92), and CI (57). Thus, the optimal proportion of populations to be sampled for conservation purposes from each geographical group can be expressed as 0.45 (IP): 0.34 (M): 0.21 (CI).

Approximately half of the *A. edulis* populations (9/17) occupy large areas (>1 km$^2$), but only 7 populations exceed 1000 individuals (Table 3). Most of the Iberian populations show large occupation areas, population sizes, and stable or moderate habitat decline. By contrast, the Moroccan populations present smaller occupation areas, population sizes,

Table 5 **Associations between geographical and qualitative population variables (factors) and genetic diversity and rarity ($h_{Nei}$, Nei's diversity index; *Nei, 1987*; DW, frequency down-weighted marker values; $N_r$, number of rare alleles), as tested using the generalized linear model (GLM).** Geographical groups: IP, Iberian Peninsula; M, Morocco; CI, Canary Islands. All the values are indicated as mean ±SE. Different letters indicate significant differences in the multiple comparison test at P < 0.05, performed after the bootstrapped GLM.

| Factor | Level | $h_{Nei}$ | DW | $N_r$ |
|---|---|---|---|---|
| Geographical group | IP | 0.12 ± 0.01a | 3.20 ± 0.47ab | 33.66 ± 4.89a |
| | M | 0.10 ± 0.01a | 2.57 ± 0.22b | 26.00 ± 5.00b |
| | CI | 0.10 ± 0.03a | 4.52 ± 0.86a | 35.33 ± 11.86a |
| Occupation area | large | 0.12 ± 0.01a | 3.30 ± 0.37a | 35.44 ± 4.06a |
| | small | 0.09 ± 0.01b | 2.96 ± 0.46a | 24.62 ± 5.31b |
| Population size | large | 0.12 ± 0.01a | 3.29 ± 0.31a | 33.57 ± 4.33a |
| | small | 0.09 ± 0.01b | 3.03 ± 0.45a | 28.10 ± 5.11b |
| Vulnerability | stable | 0.14 ± 0.01a | 4.56 ± 0.28a | 46.66 ± 4.91a |
| | declining | 0.10 ± 0.01b | 2.91 ± 0.41b | 26.44 ± 3.99b |
| | critically declining | 0.09 ± 0.01b | 2.68 ± 0.33b | 27.60 ± 7.05b |
| Legal status | protected | 0.13 ± 0.02a | 3.55 ± 0.52a | 38.75 ± 4.97a |
| | unprotected | 0.09 ± 0.02b | 3.01 ± 0.34a | 27.77 ± 4.01b |

and usually severe habitat decline. Only four populations from the Iberian Peninsula occupy protected areas, e.g. within Special Areas of Conservation of the Natura 2000 network or Andalusia regional system of protected areas (RENPA Network), while the areas occupied by the remaining populations lack legal protection.

The generalized linear model (Table 5) revealed significant influence for most of the geographic and population variables on the main genetic diversity and rarity parameters. Geographically, the Iberian Peninsula and Canary Islands accounted for higher genetic diversity than did Moroccan populations. Also, as expected, a significantly higher genetic diversity and rarity (Nei's diversity index, frequency down-weighted marker values, and number of rare alleles) was found in populations occupying larger areas, with higher numbers of individuals, stable populations, and locations in protected areas.

## DISCUSSION

### Genetic variability and structure

Although we are aware that AFLP-based estimates of the level of genetic variation are difficult to compare across studies (*Nybom, 2004*), the genetic-variation levels when standardizing sample size by population (i.e. indicating that relative differences in population diversity are not an artefact of the sampling effort) in *A. edulis* appear to approach those found in another annual species, *Hypochaeris salzmanniana* (*Ortiz et al., 2007*), which has a comparable distribution area (south-western Spain and Atlantic coast of Morocco). The diversity levels found are also comparable to those of other Mediterranean perennial herbs (*Edraianthus serpyllifolius* and *E. pumilio*; *Surina, Schönswetter & Schneeweiss, 2011*) belonging to *Astragalus* (*A. cremnophylax*; *Travis, Manchinski & Keim, 1996*), or even long-lived western Mediterranean trees

(*Juniperus thurifera*, *Terrab et al., 2008*). Nevertheless, AFLPs have relatively low genetic diversity in *A. edulis* populations, compared to that of the Iberian narrow endemic steppe shrubs *Boleum asperum* (*Pérez-Collazos, Segarra-Moragues & Catalán, 2008*) and *Vella pseudocytisus* subsp. *paui* (*Pérez-Collazos & Catalán, 2006*).

Diversity as well as rarity values are particularly useful when used to compare populations or geographic areas occupied by the study species. In *A. edulis* the maximum diversity and rarity values within the Iberian distribution range correspond to the most central populations (AE4 and AE5), and within Morocco the AE8 and AE9 populations (Table 3; Fig. 1). Contrarily, on the easternmost edge of the distribution area of the species some of the lowest diversity and rarity values were found, i.e. AE6 (IP) and AE7 (M). The central parts of the Iberian distribution of this species may represent a long-term in situ survival area. By contrast, the easternmost Iberian population AE6 could be the result of a single dispersal event, the extremely low genetic-diversity and rarity values indicating a genetic bottleneck. Within Morocco AE8 is a large population (several hundred individuals) and could have acted as a source area, as confirmed also by the NJ analysis (Fig. 2A). Meanwhile, AE7, with less than 20 individuals, could also have resulted from a single dispersal event. This hypothetical fine-scale west to east colonization pattern described for the Iberian Peninsula parallels that observed in Morocco and the low diversity and rarity values found in the easternmost Iberian and Moroccan sampling sites (AE6–AE7) may indicate that the eastward colonization history of the species in these areas might have been affected by founder effects and genetic bottleneck. This mode of peripheral founder events in small populations may be key in the future genetic differentiation of populations, as described for other plant species (e.g. *Tremetsberger et al., 2003*; *Pérez-Collazos, Segarra-Moragues & Catalán, 2008*). In both the Iberian Peninsula and Morocco, aridity is higher eastwards, which on one hand may hamper future survival of these easternmost populations but, on the other hand, may promote new genetic variants as a response to environmental selection pressure.

In the Canary Islands, diversity and rarity reached their highest levels in AE16 (Fuerteventura), and their lowest levels in AE15 (Lanzarote). Considering that both islands emerged as a single proto-island and remained together as recently as the late Pleistocene (*Fernández-Palacios et al., 2011*), the current *A. edulis* distribution could be the product of an ancient long-distance dispersal event, a recent long-distance dispersal event, or the result of range fragmentation. The observed diversity and rarity values seem to favour the hypothesis of a rather recent long-distance dispersal event from Fuerteventura to Lanzarote. In any case, AE15, as well as AE7 and AE6, had been affected by founder effects and genetic bottlenecks probably related to genetic drift.

The overall AMOVA analysis led to the conclusion that most of the overall genetic variation of the species could be attributed to intrapopulational (inter-individual) variability, while a smaller percentage of the total variation appeared among populations (Table 4). Comparing our findings with those resulting with AFLPs for other species from the western Mediterranean, either with similar distribution areas (*Ortiz et al., 2007*;

*Terrab et al., 2008*), or Iberian narrow endemic steppe plants (*Pérez-Collazos & Catalán, 2006*; *Pérez-Collazos, Segarra-Moragues & Catalán, 2008*), we detected similar patterns and divergence levels. Also similar patterns were found for the tree *J. thurifera*, which shows a wider distribution area, and surprisingly they also parallel those shown by the perennial shrubs *B. asperum* and *V. pseudocytisus* ssp. *paui*, which are very narrow endemics from NE Spain. It is well known that long-lived and outcrossing species retain most of their genetic variability within populations and, by contrast, annual and/or selfing taxa allocate most of the genetic variability among populations (*Nybom, 2004*). Nevertheless, we found similar high levels of within-population diversity for the annual *A. edulis* than for the perennials *J. thurifera, B. asperum*, and *V. pseucocytisus* ssp. *paui*, while for the annual herb *H. salzmanniana* the levels of inter-individual (within population) genetic variability are significantly lower (*Ortiz et al., 2007*). These data support the idea that the levels of intrapopulation genetic diversity are relatively high for an annual species, perhaps facilitating the preservation of the gene pool of the species and, therefore, of the evolutionary processes that generate and maintain it.

## Designing conservation strategies: selection of RGUCs

*Astragalus edulis* has a relatively high number of populations and number of individuals (at least in the large Spanish core), hampering the protection in situ of the entire distribution range of the species, and thus populations need to be identified to apply conservation measures. To select the populations deserving protection, by means of RGUCs, we propose the consideration of factors that could have influenced the evolutionary history of the species lineages (*Frankham, Ballou & Briscoe, 2009*). The selection of RGUCs has enabled the estimation of the number of populations that should be targeted to sample 99.9% of the total genetic diversity of *A. edulis*. This approach helps to select particular populations that should be prioritized because they have a singular allelic composition. The probabilities of rare-allele loss indicate that the proportions that should be preserved from each geographical group should be 0.45(IP):0.34(M):0.21(CI). Considering the diversity and rarity values found for each population based on AFLP data and also this optimal proportion of populations to be sampled for conservation purposes from each geographical group, we would initially recommend the priority selection of populations AE1, AE4 and AE5 (IP), AE8 and AE9 (M) and AE16 (CI). Nevertheless, linking genetic diversity and rarity with qualitative population and conservation features, we have found that *Astragalus eduli*s exhibit a significantly higher genetic diversity and rarity in populations occupying larger areas, with higher numbers of individuals, stable populations, and locations in protected areas.
That is the case of populations AE4, AE5 but not of populations AE1,
AE9 and AE16.

This selection of RGCUs based on AFLP data and population parameters could be complemented with the available information on haplotypes. The presence of endemic haplotypes in the three main geographical groups suggests an impact of the biogeographic barriers in the study area (Atlantic Ocean, Atlas Mountains, Alboran Sea) in shaping *A. edulis* genetic diversity and divergence. Haplotypes endemic to restricted areas

represent singular genetic variants that may have evolved separately from each other and, therefore, they deserve particular conservation efforts. Within the Iberian distribution range of the species, populations AE4 and AE5 show maximum diversity and rarity values and their sampling may warrant conservation of the Iberian endemic haplotypes III and V, apart from the widely distributed haplotypes I and IV (Table 3; Fig. 3). The selection of AE1, the Iberian population with the next highest singularity value, would additionally contribute to the conservation of the endemic haplotype V. Within the Canary Islands, population AE16 registers comparatively the highest values of singularity and diversity; moreover, the selection of AE16 for conservation purposes would warrant the conservation of haplotype VII, which is endemic to these islands. Within Morocco, populations AE8 and AE9 have comparatively the highest values of singularity and diversity, but haplotypes endemic to N Africa −II and VI, which are present in populations AE11 and AE10, respectively–would not be represented by the selection of AE8 and AE9. The protection of populations AE11 and AE10 would also be highly desirable, because in this case the evolutionary history based on the cpDNA of *A. edulis* in this geographic area would also be taken into account. Given that the Moroccan populations of this species show medium levels of genetic diversity and rarity (considering the overall values of *A. edulis*), our final decision on which particular populations from N Africa deserve priority for conservation would probably be more accurate if based on the consideration of these rare or restricted haplotypes. From this perspective, AE10 and AE11 could be prioritized over AE8 or AE9, although this decision should be taken with care given that our sampling may be low despite the results obtained from Dixon's test. The protection of large populations and smaller dispersed patches usually help preserve genetic integrity and diversity (*Alexander, Liston & Popovich, 2004*), but some selected RGUCs for *A. edulis* have small occupation areas and population sizes, and are critically vulnerable.

Several conservation measures could be implemented for the populations selected, e.g. studies to gather data on spatial distribution, population-size fluctuations, habitat quality, and fitness trends (*Morris & Doak, 2002*), reinforcement of the smallest populations, and ex situ conservation in seed banks (*Peñas, 2004*). Indeed, in order to preserve *Astragalus edulis* at long-term, including the evolutionary potential of its populations, are needed ex situ collections (e.g. botanical gardens and seed banks; *Guerrant, Havens & Maunder, 2004*) combined with any real in situ conservation value (*Cavender et al., 2015*).

The identification of highly representative populations based on genetic data is essential to design appropriate conservation guidelines, especially because this species is listed in a threat UICN category. In biological conservation it is useful to combine molecular data with additional environmental, ecological, and biological data sets in multidisciplinary approaches (*Habel et al., 2015*). The method followed here to choose RGUCs draws not only on the approach of other authors (*Ciofi et al., 1999*; *Pérez-Collazos, Segarra-Moragues & Catalán, 2008*; *Segarra-Moragues & Catalán, 2010*), but also on complementary phylogeographic, population, and ecological data. Therefore, could be more comprehensive and also perhaps more useful for management efforts that should prioritize populations to preserve the evolutionary potential of the species (*Rumeu et al., 2014*).

## ACKNOWLEDGEMENTS

We are grateful to L.M. Muñoz, J.A. Elena, S. Andrés, X. Giráldez, B. Benito, C. Oyonarte, and A. Abad for their help during field work. We thank A. Abad also for helping with the lab work. We are also grateful to Andreas Tribsch for advising on laboratory experiments, and to D. Nesbitt for the English-language revision.

### Funding

This work has been financed by the Spanish Ministerio de Ciencia e Innovación through the projects CGL2012-32574 and REN2003-09427, as well as by the Andalusian Consejería de Innovación, Ciencia y Tecnología through the project RNM1067. The funders had no role in study design, data collection and analysis, decision to publish, or preparation of the manuscript.

### Grant Disclosures

The following grant information was disclosed by the authors:
Spanish Ministerio de Ciencia e Innovación: CGL2012-32574 and REN2003-09427.
Andalusian Consejería de Innovación, Ciencia y Tecnología: RNM1067.

### Competing Interests

The authors declare that they have no competing interests.

### Author Contributions

- Julio Peñas conceived and designed the experiments, performed the experiments, analyzed the data, wrote the paper, reviewed drafts of the paper.
- Sara Barrios performed the experiments, analyzed the data, contributed reagents/materials/analysis tools, wrote the paper, prepared figures and/or tables.
- Javier Bobo-Pinilla performed the experiments, analyzed the data, contributed reagents/materials/analysis tools, prepared figures and/or tables, reviewed drafts of the paper.
- Juan Lorite analyzed the data, reviewed drafts of the paper.
- M. Montserrat Martínez-Ortega conceived and designed the experiments, performed the experiments, analyzed the data, wrote the paper, reviewed drafts of the paper.

### Data Deposition

The research in this article did not generate any raw data.

### Supplemental Information

Supplemental information for this article can be found online at http://dx.doi.org/10.7717/peerj.1474#supplemental-information.

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
