# Peer review of "Designing conservation strategies to preserve the genetic diversity of Astragalus edulis Bunge, an endangered species from western Mediterranean region"

_PeerJ, doi:10.7717/peerj.1474_

## Round 0.1 · original submission · Major Revisions

Authors present their work on conservation strategies to preserve the genetic diversity of Astragalus edulis Bunge, an endangered species from western Mediterranean region. This study is useful, however there is need of several improvements so that manuscript quality is of the standard. Hence, I recommend the authors change the manuscript as suggested by the reviewers.

Reviewer 1 ·

Basic reporting

No comments

Experimental design

No comments

Validity of the findings

No comments

Additional comments

This study investigated the diversity levels of the endangered Astragalus edulis. The main topic of this paper is quite interesting because is an example of the use of genetic data for designing conservation strategies. In general the presentation of their study is good, although some more intense efforts at organization would greatly aid the reader in following the multiple angles the authors pursue. In my opinion, the manuscript would benefit from some clarifications as pointed.

Reviewer 2 ·

Basic reporting

The files have problems in the numbering of the lines. See lines: 146-147 and 228-229.
The presentation of the figures could also be improved:

Experimental design

No Comments

Validity of the findings

No Comments

Additional comments

Designing conservation strategies to preserve the genetic diversity of Astragalus edulis Bunge, an endangered species from western Mediterranean region. Peñas et al. used molecular tools to quantify and identify which populations should be prioritized to better represent the genetic singularity and geographic variability of A. edulis for conservation. This paper addresses an area of basic importance in biology conservation, because it expected to shed light on ways to conserve biodiversity of endangered species, so I was very excited to read it. The authors found that three populations from the Iberian Peninsula, two from Morocco, and one from the Canary Islands represent the total genetic diversity of the species and the rarest allelic variation. This study therefore has the potential to make a really valuable contribution to the literature. Despite my overall positive opinion of the work and for what the authors are trying to do, I have some concerns about it that should be addressed before it is suitable for publication.
The files have problems in the numbering of the lines. See lines: 146-147 and 228-229. My remarks refer to numbering in the PDF file.

Abstract:
I think it would be interesting to identify here the molecular markers used (AFLPs and plastid spacers). This helps to ensure it appears higher in the results returned by search engines.
Alternatively this information can be added in the keywords

L. 107: "The reproductive biology of the species is poorly known..."
Do you know the pollination syndrome? Can pollination be achieved by self-pollination or cross-pollination? If you know any of these answers, please add in the text. This can help in interpreting the data.

L. 124-127: I would like to know more information of those populations that authors chose.
“samples were collected at distances greater than 5 m“. What were the minimum distances between populations? Please simply mention it.

L.135: "with modifications" Please, specify. This is important for the reproducibility of our results.

L.146-147: trnS-trnG, rpoB-trnC, according Shaw et al. 2005.
Using the same protocols? Please mention it.
"These regions showed the highest variability of 23 surveyed regions and were used to analyse a total of 61 individuals of A. Edulis". How do you know? For this, you would have to analyze the 61 individuals for all markers. Please clarify it. Why only 61? Well seems like you must have an explanation for that.

L. 249:
"The genetic variability within A. edulis was remarkably low (26 cpDNA regions tested), and all the mutations together defined a total of 7 haplotypes”. How many individuals were tested? Specify the number of mutations: Indels? Substitutions?...
I suggest the use of the Arlequin software to calculate: Number of Observed transversions, Number of substitutions, Number of Observed indels, Number of polymorphic sites, Nucleotide diversity and gene diversity, Number of observed transversions, Number of substitutions, Number of observed indels, Number of polymorphic sites, Nucleotide diversity and gene diversity.

L. 311:
“Contrarily, the lowest diversity and rarity values appeared on the easternmost edge of the distribution area of the species, i.e. AE6 (IP) and AE7 (M).”
AE10 and AE14 also exhibit low values of diversity and rarity values. Why these populations are not mentioned? Are the genetic diversity and rarity of AE6 and AE7 significantly different from those of other populations?

L. 366-369:
“That is the case of populations AE4, AE5 and AE8, but not of populations AE1, AE9 and AE16.”
According to table 2, AE8 is unprotected.

L.395-396:
I would be more careful here. An analysis to justify this decision was not performed.
In addition, a sampling of 61 individuals may not be suitable. These haplotypes (II and VI) could have been sampled in other populations with the same sample size used for AFLP.

Figures:
The presentation of the figures could also be improved:
Fig 1. Please, show the shoreline (or use a darker gray). I barely can identify the continent limits.
Please, add a metric scale.

---

## Round 0.2 · accepted · Accept

The reviewers agreed for the acceptance of the current version of the manuscript, this manuscript is officially accepted now. Congratulations to authors for their contribution.

Reviewer 2 ·

Basic reporting

No Comments

Experimental design

No Comments

Validity of the findings

No Comments

Additional comments

The authors have fully addressed the points raised. Now, this work is worth of publication at its current form.